# Ethics in healthcare: Knowledge, attitude and practices of nurses in the Cape Coast Metropolis of Ghana

**Patience Asare[1], Edward W. Ansah[1], Francis Sambah[1,2]***

**1** Department of Health, Physical Education and Recreation, University of Cape Coast, Cape Coast, Ghana,
**2** College of Public Health, Medical and Veterinary Sciences, James Cook University, Townsville, Queensland, Australia

* francis.sambah@stu.ucc.edu.gh

## Abstract

### Background

Nursing is a profession that care for personal and private aspects of people's lives. Therefore, nurses need to know the basic ethical aspects of nursing which is integral in nursing practices. The purpose of the study was to describe the ethical knowledge, attitude and practice of nurses in the Cape Coast Metropolis of Ghana.

### Method

A cross-section design was used to collect data from 264 nurses in three selected healthcare facilities in the Metropolis. A structured questionnaire was administered to all the categories of these nurses in the selected facilities. Frequency counts and multiple regression statistics were used to analyze the data.

### Results

The results show 78% of nurses possess good ethical knowledge, 84% had a positive attitude, while 98% had good ethical practices. The results further show that nurses' professional rank [F (1, 259), 2.35, p = .02] and academic qualification [F (1, 259), 2.67, p = .008] were significant predictors of their ethical knowledge and attitude, respectively.

### Conclusion

Inadequate resources, poor set up of working areas and understaffing are the major barriers limiting the practice of good ethical standards among the nurses. The Regional Health Directorate, the Ministry of Health and the Managers in charge of the health facilities need to work together to eliminate these barriers as they have the potential to negatively impact quality healthcare delivery in the Metropolis.

**Data Availability Statement:** All relevant data are publicly available via DOI 10.17605/OSF.IO/SXMGU (https://osf.io/sxmgu/).

**Funding:** The authors received no specific funding for this work.

**Competing interests:** The authors have declared
that no competing interests exist.

## Introduction

Ethical issues in healthcare and nursing practice are becoming more complex with medical
advances, and the increased dynamics of the healthcare system [1]. Ethics in healthcare is the
standards or principles of moral judgment or actions which provide a methodical system in
differentiating right from wrong based on certain beliefs [2]. And because nurses spend a lot
time with patients and patient families, the practices of nurses in relation to their clients
become paramount (Kieft et al., 2014). Thus, the first step for a nurse to make ethical decisions
is to identify and understand the ethical issues surrounding patient care. Ethical nursing prac-
tice involves core ethical responsibilities that nurses are expected to uphold, which date back
since 1953 [3]. Ethical codes in nursing, serve not only to define nursing "profession" but also
outline primary duties, responsibilities and obligations towards their clients [4]. In that case,
professional code of ethics for nurses serves as a guide for carrying out nursing responsibilities
in a manner consistent with quality care, ethical obligations of the profession, acceptance of
the rights of individuals, and for patients' safety [5]. Moreover, individual dignity needs to be
respected regardless of who is receiving the care, poor, rich, black, white, Christian, Muslim,
male or female, a child or an adult [6], or a person of any gender.

It is important that nurses acquire and practice with the necessary knowledge of ethics to
impact positively on the lives of their patients, patient family, and the society [7]. However,
various studies have shown nurses' weaknesses in the knowledge of ethics and its application
in practice towards patients and their families [8, 9]. For instance, it has been reported that
nurses' approaches to ethical problems in Ghana do not always meet expectations of the inter-
national code of practice [7]. Accordingly, nurses' ethical practices are being informed by local
ethical practices which are related to the institutional setting and cultural environment in the
country, Ghana [8]. Unfortunately, while some cultural values are complementing to the inter-
national best code of ethics and principles, many others were conflicting [7].

In Ghana, the Nursing and Midwifery Council (NMC), a statutory body responsible for
nursing and midwifery professions, has a code of ethics. The code of ethics is meant to inform
nurses and midwives of the best standard required in the exercise of their professional
accountability, to inform the public, other professionals, and employers of the standard of pro-
fessional conduct expected of registered nurses and midwives in the discharge of their duties
at the various healthcare facilities [6]. Thus, demonstration of any unethical conduct by nurses
on duty of care would mostly come as a result of their level of knowledge, attitude and many
other factors such as institutional and social settings [10]. For instance, unethical behaviors of
nurses are believed to be related to insufficient personnel, poor working conditions, excessive
workload, lack of supervision, and minimal level of in-service training [10]. Evidence further
suggests that majority of nurses do not acknowledge ethical issues at work, for which they
exhibit unethical behaviors in their practice [11]. Furthermore, personal variable such as age,
gender, level of education, professional qualification, marital status, years of experience, rank,
ward setting, training, and institution could have significant influence on nurses' level of
knowledge, attitude, and practice of ethics in their professional duties [11–13]. These practices
do not only negatively affect the health outcomes of patients under care, they equally reduce
the professional image of nurses, their facilities, and the nursing profession [10, 13].

The aim of nursing profession is to provide the highest possible standard of care to patients,
which requires that a high standard of professional behavior is exhibited [14]. Studies have
shown that nurses lack ethical knowledge, portray negative ethical attitudes, and failed to
apply their ethical codes in caring for their patients [9]. For example, media report revealed
that in Ghana, nurses in the Cape Coast Metropolis have fallen short of applying ethical knowl-
edge in their duties [6]. Moreover, it has been observed that at-duty-nurses insult their patients

for expression of labor pain during delivery [7, 15]. For example, women who attended antenatal clinics complained about how nurses embarrassed them because of the women's failure to attend antenatal appointments on schedule and their inability to afford some of the items required for delivery [16]. Beyond these, nurses are often described as unfriendly, stubborn, disrespectful, saucy, cruel, callous, and inhuman to their clients [17, 18]. Despite these growing concerns and incidences of unethical practices of nurses in health facilities in the Cape Coast Metropolis, there is a paucity of literature on the extent of such behavior and factors that might be influencing such practices. Hence, this study aimed to explore knowledge, attitude, and practices of ethics in nursing care among nurses in the Cape Coast Metropolis.

## Methods

This was a cross-sectional survey involving 264 nurses from three healthcare facilities within the Cape Coast Metropolis (i.e. Cape Coast Teaching Hospital [CCTH], Municipal Hospital [MH] and University of Cape Coast Hospital [UCCH]. The data were collected between September and December 2016, in the three major healthcare facilities within the Cape Coast Municipality. This study applied a census, targeting all nurses, but those nurses on annual leave, study leave, sick leave, or on special assignment out of the facilities could not take part in the study. Thus, we are of the view that these group of nurses represent the general nursing population in the Cape Coast Municipality, because the three healthcare facilities have all the categories of nurses. Moreover, we recorded a return rate of approximately 81% (264 out of a total of 326). Thus, all the 326 nurses working in the municipality hospitals were included in the study.

### Dependent measure

We used a 32-items questionnaire, developed by the researchers, to collect data. The instrument was developed from previous empirical literature on ethics and the code of ethics of Nurses and Midwives Council (NMC) of Ghana. The questionnaire solicited information on knowledge, attitude, and practice of ethics, and was in two sections, A and B. Section A was made up of six multiple choice items which solicited nurses' background characteristics such as sex, age, professional qualification and rank, academic qualification, and years of working experience. With 24 items, the section B measured knowledge, attitude, and practice of ethics in nursing practice. This section also collected information on the nurses' sources of knowledge (Q29) and barriers (Q30) to their ethical practice. The section was subcategorized into I, II and III, with I measuring knowledge about ethical practices (Q7-Q16), which the participants responded to as yes = 3; no = 2; do not know = 1. Subsection II contained two items (Q17-Q22) that solicited information about nurses' ethical attitudes, where participants responded to as strongly disagree = 1; disagree = 2; agree = 3 and strongly agree = 4. Furthermore, subsection III (Q23-Q28) measured the extent of ethical practice among nurses, with the response categories of never = 1, seldom = 2, occasionally = 3 and always = 4. Section B yielded separate scores for knowledge, attitude, and practice of ethics by nurses. The high scores or arithmetic means reflect maximum performance on that particular construct (i.e. knowledge, attitude and practice).

Copies of the developed questionnaire were initially given to five university level and four diploma level nursing students to assess. The reviewed questionnaire was then given to two nursing tutors and two lecturers for their expert inputs. To pilot test the questionnaire, we further administered it to 79 conveniently sampled nurses from Asankrangwa Catholic Hospital and Samartex Hospital at Samreboi both in the Wasa Amenfi West District in the Western Region of Ghana. This group of nurses did not take part in the main study. Instrument yielded

an internal consistency reliability value of 0.73, based on the pilot test and 0.78 from the main study.

The study protocols were approved by the Institutional Review Board (IRB) of the University of Cape Coast, Ghana (ID: UCCIRB/CES/2015/1), and permission was granted by the heads of administration and unit heads of the three healthcare facilities. To increase anonymity and confidentiality of the information provided by the nurses, copies of the instrument were put in unsealed envelopes and deposited with the ward in-charges who did the onward distribution to the nurses. Using the instruction on the introductory page of the questionnaire, we directed the nurses to seal the envelope containing the filled questionnaire before returning it to their in-charges. We utilized the ward in-charges in the data collection because they work daily with the nurses. The nurses were given three days to complete the instrument and returned same to their in-charges for the researchers to retrieve. We also attached an informed consent form to the instrument which every participant was supposed to sign.

## Data analysis

Frequency and percentage counts were used to determine nurses' level of knowledge, attitude and practice, and barriers to professional ethical conduct at work. Knowledge was measured initially as *yes (3)*, *no (2)* and *I don't know (1)* and was later categorized into two, yes and *no because no* and *I don't know* were put together. Thus, from the 10 items measuring knowledge, the scores range from lowest of 10 to highest of 30, and participant must score 21 or higher to be categorized as knowledgeable, between 11-and 20 was considered moderate level of knowledge, and between 1 and 10 as low in ethical knowledge. Moreover, nurses' attitude towards ethical issues at work was recategorized as positive or negative, from the initial four-point scale. Attitude was measured with six items (Strongly Disagree = 1 to Strongly Agree = 4), and the higher the score, the more positive the attitude of the participant is towards nursing ethical issues. And that, a nurse must have scored a composite score of 13 to be deemed to have positive attitude.

Furthermore, the extent of ethical practice was classified as never, seldom, occasionally and always (1 to 4), and the higher the composite score, the better the practices. In that case, a score between 13 and 24 is classified as adequate practice and that between 1 and 12 inadequate. Moreover, barriers to the practice of ethics among nurses were analysed and reported with the various items listed on the instrument using frequency and percentage analysis.

We further applied multiple regression analyses to determine the extent to which background characteristics of the nurses influence their knowledge, attitudes, and the extent of ethical practice at their various units of work. The constructs- knowledge (measured with 10 items), attitude (six items) and practice (six items) were aggregated to for composite numeric or continuous scores that satisfied regression analysis. Thus, we did run separate regression analysis for each of these variables against the demographic variables which have different levels of measurements. Statistical significance levels were determined at p value<0.05 [19, 20]. SPSS version 20.0 software was used to run all the statistical analyses.

## Results

The majority of the participants (n = 194; 74%), were females, with only 26% (n = 70) being males. Sixty seven percent (n = 178) of the nurses were between ages 21 and 30, 24% (n = 62) were between 31 and 40 years, and nine percent (n = 24) were 41 years or above (See Table 1 for the rest).

The results show that 77.7% (n = 205) of the nurses recorded good knowledge, 66.6% (n = 44) moderate, whereas 5.7% (n = 15) low. Again, majority, 83.7% (n = 221), of the nurses

**Table 1. Demographic characteristics, ethical knowledge, attitudes, practices and barriers to ethical practice by nurses.**

|  | No. | % |
|---|---|---|
| Gender |  |  |
| Male | 70 | 26 |
| Female | 194 | 74 |
| Age |  |  |
| 21–30 | 178 | 67 |
| 31–40 | 62 | 24 |
| 41+ | 24 | 09 |
| Education |  |  |
| Diploma | 138 | 52 |
| Certificate | 71 | 27 |
| Bachelor or higher | 55 | 21 |
| Categories |  |  |
| Registered General Nurse | 138 | 52 |
| Health Assistant Clinical | 55 | 21 |
| CHN/RCN/RMN | 41 | 16 |
| Registered Midwives | 30 | 11 |
| Working Experience |  |  |
| 5 Yrs or Less | 177 | 67 |
| More Than 5 Yrs | 87 | 33 |
| Knowledge |  |  |
| Good | 205 | 77.7 |
| Moderate | 44 | 16.6 |
| Poor | 15 | 5.7 |
| Attitude |  |  |
| Positive | 221 | 83.7 |
| Negative | 43 | 16.3 |
| Practice |  |  |
| Adequate | 258 | 97.7 |
| Inadequate | 6 | 2.3 |
| Barriers |  |  |
| Inadequate resources | 187 | 70.8 |
| Poor set up of working area | 165 | 62.5 |
| Understaffing | 159 | 60.2 |
| Inadequate education about the code of ethics | 136 | 51.5 |
| Conflicts between nurses and doctors | 111 | 42.0 |
| Working with unethical colleagues | 111 | 42.0 |
| Overcrowding | 108 | 40.9 |
| Lack of space | 106 | 40.2 |
| Working with incompetent colleagues | 99 | 37.5 |
| Conflict with superiors | 92 | 34.8 |
| **Total** | **264** | **100** |

Note: CHN = Community Health Nurse; RCN = Registered Community Nurse; RMN = Registered Mental Nurse.

showed positive attitude, while 16.3% (n = 43) reported negative attitude towards ethics of their profession. Furthermore, the results reveal that 98% (n = 258) of the nurses recorded high practice of ethical standards at work, while 2.3% (n = 6) did not (see Table 1). The results

**Table 2. Influence of sex, age, academic qualification and rank of nurses on the level of their ethical knowledge.**

|  | B | R | R² | β | t | p-value |
|---|---|---|---|---|---|---|
| Constant | 6.69 | .30 | .09 |  | 17.41 | .001 |
| Sex | .00 |  |  | .00 | .02 | .985 |
| Age | .00 |  |  | .01 | .16 | .874 |
| Academic qualification | .17 |  |  | .12 | 1.57 | .119 |
| Rank | .48 |  |  | .21 | 2.35 | .020 |

*df* (4, 259); *F* (17.41)

on barriers to ethical practices reveal that more than half (71%, 63%, and 60%) of the nurses indicated that inadequate resources, poor set up of working areas and understaffing (among many others) prevent them from practicing good ethics (see Table 1).

Multiple regression analysis was calculated to predict nurses' ethical knowledge using sex, age, rank and academic qualification as predictors. The model revealed a significant prediction, $F$ (4, 259), 17.41, $p$ = .001, accounting for 9% of the variance in nurses' knowledge on ethics. Specifically, participants' professional rank was the only independent variable significantly predicting nurses' ethical knowledge $F$ (1, 259), 2.35, $p$ = .02, and accounted for 48% of their knowledge on ethics (see Table 2).

The second multiple regression analysis also indicated that the model significantly predicted nurses' attitude to ethics, $F$ (5, 258), 18.02, $p$ = .001, accounting for a 6% variation in nurses' attitudes towards ethics. The age of the participants was not included in this analysis because of the high correlations between age and working experience ($r$ = .91). The analysis further indicated that nurses' academic qualification was the only independent variable predicting their attitude towards ethical practice, $F$ (1, 259), 2.67, $p$ = .008, accounting for about 27% of the change in their attitude towards ethical practice (see Table 3).

The third regression model also indicated a significant prediction of nurses' ethical practice, $F$ (5, 258), 24.23, $p$ = .001, accounting for 2% variation in nurses' ethical practice. However, we did not include nurses' academic qualification in this analysis since academic qualification and professional qualification recorded high correlation coefficient ($r$ = .70). However, none of these predictor variables was independently statistically significant in determining nurses' ethical practice (see Table 4).

## Discussion

The focus of this study was to describe the knowledge, attitude, and ethical practices among nurses in the Cape Coast Metropolis of Ghana, and determine the extent to which

**Table 3. Influence of sex, academic qualification, professional qualification, rank and working experience of nurses on their attitude towards ethical practice.**

|  | B | R | R² | β | t | p-value |
|---|---|---|---|---|---|---|
| Constant | 17.41 | .25 | .06 |  | 18.02 | .001 |
| Sex | −.23 |  |  | −.03 | −.55 | .585 |
| Academic qualification | 1.12 |  |  | .27 | 2.67 | .008 |
| Professional qualification | .01 |  |  | .01 | .11 | .916 |
| Rank | −.69 |  |  | −.11 | −1.19 | .236 |
| Working experience | .03 |  |  | .06 | .79 | 427 |

*df* (5, 258); *F* (18.02)

**Table 4. Influence of sex, academic qualification, professional qualification, rank and working experience of nurses on the extent of their ethical practice.**

|  | B | R | $R^2$ | β | t | p-value |
|---|---|---|---|---|---|---|
| Constant | 21.82 | .15 | .20 |  | 24.23 | .001 |
| Sex | .05 |  |  | .15 | 1.80 | .073 |
| Age | −.13 |  |  | −.02 | −.37 | .712 |
| Academic qualification | .12 |  |  | .11 | 1.22 | .226 |
| Rank | −.25 |  |  | −.05 | −.53 | 596 |

*df* (5, 258); *F*(24.23)

demographic factors of the nurses influence their knowledge, attitude and practice of ethics in the nursing profession. We observed a good level of ethical knowledge among the nurses. First, we are of the view that the nurses in this study recognize ethical problems in the course of their care practice because of seminars and workshops they attended during the period of practice, as has been reported in other studies [7, 21]. Further, ethics is emphasized across all levels of nursing education globally and in Ghana [6, 22]. Besides, every professional group has established a code of ethics and standards, normally drawn from the ICN, so, it does not matter where the nurse is trained, they are expected to adhere to those ethics [23], which agrees with previous studies in Turkey [10], Spain [21], Australia [24], Italy [25], Iran [26], and Hong Kong [27]. Conversely, other previous findings from Northern India [8], Ethiopia [13], Germany [28], and Nigeria [29] revealed a low to moderate level of ethical knowledge. The reason for these dissimilarities could be attributed to methodological limitations. For example, while the earlier studies [8, 13] used nurses and other healthcare workers at primary healthcare facility levels, and [28] from national population surveys, our participants are nurses from secondary and tertiary facilities who are not necessarily representative of all nurses in Ghana. Though, knowledge alone does lead to practice, having a sufficient ethical knowledge may positively influence the professional conduct of nurses, which has been a great public concern in Ghana in recent times [17, 18].

The finding again indicates a good attitude of nurses towards ethical practice. We reason here that participants are applying their ethical knowledge to practice, which may be influencing their attitude. Moreover, many of these nurses come in contact with various ethical issues in their practices, evidence which were reported in Oceania [11], Africa [12, 13], Asia [27], and Europe [30, 31]. Healthcare delivery in the Cape Coast Metropolis of Ghana may be improved through positive attitude of nurses because it is likely to enhance health outcome of patients since negative attitudes from nurses could compromise patient's recovery [10, 13]. For instance, a study by Norbergh et al. [31] indicated that positive attitude of healthcare professionals is important for the wellbeing and quality of care they give their patients because such attitudes enhance patients' satisfaction, health, and psychological wellbeing.

We further observed adequate ethical practice among our participants. The reason may be because participants do well understand the issues in ethics that they were taught during their training. Moreover, these nurses may have encountered ethical issues which could inform their adequate practice. Similarly, other studies conducted in Africa [12, 13] among nurse participants found nurses' ethical practices to be adequate or good. Ethical practice is essential not only to nursing practice but also to patients [1, 4]. Moreover, practices deemed unethical can compromise the health of the client, and cause a lawsuit against the practitioner, resulting in the revocation of license and/or imprisonment. This is because flouting the NMC ethical code and causing harm to a client could be punishable by law in Ghana [6]. However, studies from Nigeria [32] and West Indies [33] found inadequate ethical practice among nurses. The

disparities in the findings may be because of the vast time period between these studies and the current one. For example, the study from Nigeria [32] is 10 years earlier to the current study. This time interval is enough in an evolving industry like healthcare for some ethical transformations and improvement to take place among professionals including nurses.

There is also an indication that inadequate resources, poor set up of the working area and understaffing are major barriers to ethical practices among these nurses. Plausibly, most of these nurses do not get the necessary or basic resources (i.e. consumables, conducive working environment to ensure confidentiality and privacy of clients) which is demotivation to adherence to ethical practices. In low-middle-income countries such as Ghana, healthcare delivery is bedeviled with several barriers, including space, understaffing and lack of or inadequate provision of resources, to enhance caregiving to patients [18, 34]. For example, evidence [34, 35] indicates that understaffing may force nurses to abandon opportunities to provide good nursing care to their patients [18]. Understaffing may also increase workload, absenteeism, turnover, work stress, burnout, and job dissatisfaction of the care practitioner [32, 34]. In this case, quality healthcare delivery may fester unprofessional conduct and become counterproductive to health outcomes of patients.

We further observed that participants' professional ranks influence their level of ethical knowledge. Perhaps, majority of the nurses in this study are at junior ranks and are likely to be young and may have completed school in not-too-distant time [21]. It is also sound to theorized that as these nurses move from one rank to another, they encounter different ethical dilemmas which may increase their knowledge of ethics, as found in Nigeria [12], New South Wales [36], and Kenya [37]. Moreover, we realized that academic qualification positively determined nurses' attitude towards ethical practice at work. Deductively, majority of the study participants attained diploma degree, who also possess a good level of knowledge and a positive attitude towards ethical practices. This is similar to a study from Ethiopia [13] which found that level of education/academic qualification influences nurses' attitude towards care of their clients. Though our study did not draw cause and effect linkage, it is sound to infer that, higher educational qualification has a propensity to positively transformed attitudes towards practice in healthcare.

This is a cross-section study, and we do not intend to infer cause and effect, generalize our findings and conclusions to the whole of Ghana or other healthcare professionals. This study further opens the discourse on nurses' ethical knowledge, attitudes, and practices during the healthcare delivery process, in the Cape Coast Metropolis of Ghana. Also, because it was a survey, the nurses may not have given us the "true picture" of their practices as may be happening at their various wards.

## Conclusions

The focus of this study was to describe the knowledge, attitude, and ethical practices among nurses in the Cape Coast Metropolis of Ghana, and determine the extent to which their demographic characteristics influence their knowledge, attitude and practices of ethics at work. Though these nurses in Cape Coast Metropolis recorded good ethical knowledge, have positive attitude towards ethical practice, and are ethically conscious in their practices, and they are faced with inadequate resources. Furthermore, professional rank and academic qualification determined the ethical knowledge and attitude of these nurses and their practices while they attempt to care for their clients.

## Recommendations

Nurses need to continually update their knowledge through seminars/workshops so they can be well-informed of ethical issues arising in their practices. Nurse managers need to do more

periodic monitoring to maintain the high image of the profession. The teaching of ethics at the training schools needs to be intensified since most of the nurses obtained such knowledge from school before entering into practice. Moreover, we recommend sanctions and penalties for nurses who are reported and confirmed to be involved in unethical practices. In addition, there is the need to provide the needed resources, increase the staff strength, and get better working space to help these nurses give the needed quality of care to their patients. Extending this study nationwide will be ideal, to get wider views and understanding of the issues studied. Also, longitudinal and or observational study or studying this phenomenon from the clients' perspectives will be of benefit to the healthcare system.

## Author Contributions

**Conceptualization:** Patience Asare.

**Data curation:** Patience Asare, Edward W. Ansah, Francis Sambah.

**Formal analysis:** Patience Asare, Edward W. Ansah.

**Funding acquisition:** Edward W. Ansah.

**Investigation:** Patience Asare, Edward W. Ansah, Francis Sambah.

**Methodology:** Patience Asare, Edward W. Ansah, Francis Sambah.

**Project administration:** Patience Asare, Edward W. Ansah, Francis Sambah.

**Resources:** Patience Asare, Edward W. Ansah, Francis Sambah.

**Software:** Patience Asare, Edward W. Ansah, Francis Sambah.

**Supervision:** Patience Asare, Edward W. Ansah, Francis Sambah.

**Validation:** Patience Asare, Edward W. Ansah, Francis Sambah.

**Visualization:** Patience Asare, Edward W. Ansah, Francis Sambah.

**Writing – original draft:** Patience Asare, Edward W. Ansah, Francis Sambah.

**Writing – review & editing:** Patience Asare, Edward W. Ansah, Francis Sambah.

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
