## [Decision Letter · Decision Letter 0]

28 Sep 2020

PONE-D-20-10854

Ethics in healthcare: Knowledge, attitude and practices of nurses in the Cape Coast Metropolis of Ghana

PLOS ONE

Dear Dr. Sambah,

Thank you for submitting your manuscript to PLOS ONE. After careful consideration, we feel that it has merit but does not fully meet PLOS ONE’s publication criteria as it currently stands. Therefore, we invite you to submit a revised version of the manuscript that addresses the points raised during the review process.

All three reviewer raise some important points and the nurse reviewers for this paper were particularly interested in the inclusion of nursing ethics, both in terms of theoretical approaches and also within professional guidelines in Ghana. An inclusion of nursing ethical theory  in your paper will really strengthen your work and we look forward to your revised paper.

We look forward to receiving your revised manuscript.

Kind regards,

Fiona Cuthill, PhD

Academic Editor

PLOS ONE

Journal Requirements:

2. In your Methods section, please provide additional information about the participant recruitment method and the demographic details of your participants. Please ensure you have provided sufficient details to replicate the analyses such as: a) the recruitment date range (month and year), b) a description of any inclusion/exclusion criteria that were applied to participant recruitment, c) a table of relevant demographic details, d) a statement as to whether your sample can be considered representative of a larger population, e) a description of how participants were recruited, and f) descriptions of where participants were recruited and where the research took place.

4.We suggest you thoroughly copyedit your manuscript for language usage, spelling, and grammar. If you do not know anyone who can help you do this, you may wish to consider employing a professional scientific editing service.  

5.We note that you have indicated that data from this study are available upon request. PLOS only allows data to be available upon request if there are legal or ethical restrictions on sharing data publicly. For more information on unacceptable data access restrictions, please see http://journals.plos.org/plosone/s/data-availability#loc-unacceptable-data-access-restrictions.

Reviewers' comments:

Reviewer's Responses to Questions

**Comments to the Author**

1. Is the manuscript technically sound, and do the data support the conclusions?

Reviewer #1: Partly

Reviewer #2: Yes

Reviewer #3: Yes

2. Has the statistical analysis been performed appropriately and rigorously? 

Reviewer #1: Yes

Reviewer #2: I Don't Know

Reviewer #3: Yes

3. Have the authors made all data underlying the findings in their manuscript fully available?

Reviewer #1: Yes

Reviewer #2: No

Reviewer #3: Yes

4. Is the manuscript presented in an intelligible fashion and written in standard English?

Reviewer #1: Yes

Reviewer #2: Yes

Reviewer #3: Yes

5. Review Comments to the Author

Reviewer #1: Review report 30th July, 2020

Summary of Review and overall impression

Ethical issues in nursing is an important area on nursing given that Nurses interact with patients at very delicate point in their lives when they are vulnerable to abuse of their rights.

However, there are important issues that need to be included.E.g the specific ethical principles, theories etc. that are of importance to nursing. The authors need to demonstrate how these principles are integrated in the International Council of Nurses guidelines and operationalized in National nurses’ regulatory body and curricular for nurses in Ghana.

There is also need to describe how the ethical guidelines for nurses relate to the specific areas of Ghana constitution

My general view is that assessment of knowledge, attitude and practice of nurses in the area of ethics is an important issue but the manuscript needs to be revised substantially before publication as follows:

Introduction: There is need for major revision to give description in the subject matter of ethics, magnitude of the problem, the negative effects and gaps in Knowledge, attitudes and practice that justify the study.

The literature review does not demonstrate logical arguments that justify the study

Methodology: There are major methodological issues of concern indicated on the specific areas (see comments in the manuscripts for detailed suggestions).

Results and Discussion: Need to be revised as indicated in the comments within the document.

Generally-there is need to revise the manuscript for logical flow.

Declaration of competing interest-I declare that I do not have any conflicting interest

The specific questions and remarks

Items Remarks

1. The study presents the results of original research √

2. Results reported have not been published elsewhere. √

3. Experiments, statistics, and other analyses are performed to a high technical standard and are described in sufficient detail -Needs revision

4. Conclusions are presented in an appropriate fashion and are supported by the data. -Need revision

5. The article is presented in an intelligible fashion and is written in Standard English. -Needs Re organization to reflect logical flow

6. The research meets all applicable standards for the ethics of experimentation and research integrity.

-Ethical issues observed not clearly explained

7. The article adheres to appropriate reporting guidelines and community standards for data availability

-Yes but needs improvement

Reviewer #2: This is an interesting and informative paper on a geographically understudied area of nursing practice. As I am a qualitative researcher, I cannot comment on the regression analysis specifically. Outside of that, the study design and methodology seem sound and the results support the conclusions, with one caveat:

I would encourage the authors to consider and discuss the difference between 1) 'ethics' in a general social sense, 2) professional ethics as specific to nursing, and 3) the particular written national codes of ethics they discuss in the paper. They should make clear that when discussing 'nurse's knowledge of ethics', they mean the third sense, namely formal written codes of conduct. I'm saying this because it could be well possible that rather than having 'no ethics at all', as the authors seem to imply, the nurses in question simply operate on a different (maybe not formal/bureaucratic) understanding of ethics, which then conflicts with professional standards. For example, their treatment of women in antenatal clinics may not be in tune with formal ethical standards at the hospital - but is it possible that it conforms to a 'local' ethics which demands that women attend the clinic and pay for the items needed there? These distinctions might make for an interesting follow-up study, but as it stands, the data does not support the claim that nurses do not have any knowledge of any ethics at all - just that they do not have knowledge of the relevant formal codes applying to their jobs.

I would also recommend to have the article professionally edited if at all possible to eliminate potential misunderstandings due to language use (e.g. p 57 ln 59 - 'couples'?).

The authors state that the original data will be made available upon request, but it appears that PLOS ONE requires data to be deposited in a public depository for the paper to be accepted. Once these points are addressed, I warmly recommend the paper for publication.

Reviewer #3: 1. Page Number - PN (3) Line Number - LN (58): - Is this issue of unethical practice among nurses a global one or just

specific to some places? Or are there other forms in other places?

2. PN (3) LN (75-79): - Where were these studies conducted? Is this a global problem or localized? You need to be specific

here. There might be a need to present existing global studies especially from the western world in relation to the

unethical nursing practices.

LN 79 - Please change to ‘Midwifery’.

3. PN (4), LN (85-86): - This sentence is not clear. How does the attitude and the practice of nurses influence how they act

and conduct their duties? Or that the nurses’ knowledge of the ethics of their profession influenced their attitudes and

practices?

LN (95-97): - Is this referring to the nurses in Ghana?

4. PN (5), LN (116): - Change to ‘bachelor’s degree’

5. PN (6), LN (144-147): - This appears confusing. Can you please elaborate on this sentence especially regarding the

evaluation by the two senior research assistants? Was this part of the analysis?

LN (149-150): This is not clear. Which of the group of nurses did not take part in the main study? Why were they not

involved in the main study? Was there a pilot test?

6. PN (10), LN (273): - Change to ‘…and in Ghana’

LN (294): Please rewrite this.

7. PN (11), LN (305-307): - Yes, this is true. However, do the nurses in Ghana or any of the African countries face lawsuits

for unethical practices? Are the people aware of their rights regarding the unethical practices of the nurses

who take care of them? Have you also considered the issues of quackery in the nursing profession, whereby people

assumed that everyone working in health facilities wearing ‘white uniforms’ are nurses? In essence, what role does

quackery play in the unethical practices observed among the nurses in Ghana?

LN (314 - 320): From your finding, and with the high percentage of ethical practices among nurses in Ghana, what could

be the explanation for these inhibiting factors raised here and the high percentage of ethical practices among the

nurses interviewed in this study?

8. PN (12), LN (332-333): - You need to expatiate more on this issue. Are diplomate nurses found to be more involved in

unethical practices as a result of their low level of education? Or what?

From this study findings, nurses who have diploma in nursing also have high level of knowledge about the ethics of

nursing profession. This appears to contradict earlier studies in relation to this.

LN (349-350): How can the findings from this study be generalized to other health care professionals? Are they also

influenced by the same factors? Are there studies that suggest that they also practice unethically? Which of the health

care professionals do you want to generalize your findings to?

LN (350-353): Further study should be encouraged to identify specific nursing specialties where there are reports of

unethical practices. There is also a need to identify the specific aspects of the nursing ethics where unethical practices

have been identified among nurses. Studies should be done to identify if there are complaints about unethical practices

among other health care professionals.

9. PN (13), LN (357-358): There should be sanctions and penalties for nurses who are reported and confirmed to be

involved in unethical practices.

6. PLOS authors have the option to publish the peer review history of their article (what does this mean?). If published, this will include your full peer review and any attached files.

Reviewer #1: No

Reviewer #2: **Yes: **Dr Steph Grohmann

Reviewer #3: No

---

## [Author Response · Author response to Decision Letter 0]

21 Apr 2021

Dear Editor and Reviewers,

On behalf of all authors, I convey our gratitude to you for the critical and constructive review that has led to the improvement of our paper entitled “Ethics in healthcare: Knowledge, attitude and practices of nurses in the Cape Coast Metropolis of Ghana”. We have revised the manuscript based on the comments raised by both the Editor and the Reviewers. We believe the manuscript has improved substantively and will be published in your reputable journal, PlosOne. All the changes have been marked yellow colour in the revised manuscript. Please address all correspondence to me via email at: francis.sambah@stu.ucc.edu.gh

Thank you

Editors Comments

1. Comment:

Please ensure that your manuscript meets PLOS ONE's style requirements, including those for file naming. The PLOS ONE style templates can be found at https://journals.plos.org/plosone/s/file?id=wjVg/PLOSOne_formatting_sample_main_body.pdf and https://journals.plos.org/plosone/s/file?id=ba62/PLOSOne_formatting_sample_title_authors_affiliations.pdf

Response: We have made the necessary corrections in conformity with the PLOS ONE’s style. 

2. Comment:

 In your Methods section, please provide additional information about the participant recruitment method and the demographic details of your participants. Please ensure you have provided sufficient details to replicate the analyses such as: a) the recruitment date range (month and year), b) a description of any inclusion/exclusion criteria that were applied to participant recruitment, c) a table of relevant demographic details, d) a statement as to whether your sample can be considered representative of a larger population, e) a description of how participants were recruited, and f) descriptions of where participants were recruited and where the research took place.

Response: We have made efforts to review the method section and included the dates for the study, inclusion and exclusion criteria. In addition, we put the demographics into a table (Table 1). 

3. Comment:

Please include additional information regarding the survey or questionnaire used in the study and ensure that you have provided sufficient details that others could replicate the analyses. For instance, if you developed a questionnaire as part of this study and it is not under a copyright more restrictive than CC-BY, please include a copy, in both the original language and English, as Supporting Information.

Response: We reworked on these, describing the questionnaire and how data was collected. 

4. Comment: 

We suggest you thoroughly copyedit your manuscript for language usage, spelling, and grammar. If you do not know anyone who can help you do this, you may wish to consider employing a professional scientific editing service. 

Response: We have made the necessary efforts to edit the manuscript and also sought a help from a professional scientific editor. 

Comment:

Upon resubmission, please provide the following: The name of the colleague or the details of the professional service that edited your manuscript. A copy of your manuscript showing your changes by either highlighting them or using track changes (uploaded as a *supporting information* file). A clean copy of the edited manuscript (uploaded as the new *manuscript* file).

Response: We have employed Dr. Jacob Owusu Sarfo and Mr. Frank Mensah for copy edition the manuscript.

 5. Comment:

We note that you have indicated that data from this study are available upon request. PLOS only allows data to be available upon request if there are legal or ethical restrictions on sharing data publicly. For more information on unacceptable data access restrictions, please see http://journals.plos.org/plosone/s/data-availability#loc-unacceptable-data-access-restrictions.

Response: The dataset supporting this manuscript has been uploaded into the Open Science Framework platform DOI 10.17605/OSF.IO/SXMGU

Comment: 

Response: The data is freely available at Open Science Framework platform via DOI 10.17605/OSF.IO/SXMGU

Response: The dataset supporting this manuscript has been uploaded into the Open Science Framework platform DOI 10.17605/OSF.IO/SXMGU

6. Comment:

PLOS requires an ORCID iD for the corresponding author in Editorial Manager on papers submitted after December 6th, 2016. Please ensure that you have an ORCID iD and that it is validated in Editorial Manager. To do this, go to ‘Update my Information’ (in the upper left-hand corner of the main menu), and click on the Fetch/Validate link next to the ORCID field. This will take you to the ORCID site and allow you to create a new iD or authenticate a pre-existing iD in Editorial Manager. Please see the following video for instructions on linking an ORCID iD to your Editorial Manager account: https://www.youtube.com/watch?v=_xcclfuvtxQ

Response: The corresponding author and one of the co-authors have added their ORCID iDs.

Reviewer #1: 

Summary of Review and overall impression

Comment:

Ethical issues in nursing is an important area on nursing given that Nurses interact with patients at very delicate point in their lives when they are vulnerable to abuse of their rights. However, there are important issues that need to be included. E.g the specific ethical principles, theories etc. that are of importance to nursing. The authors need to demonstrate how these principles are integrated in the International Council of Nurses guidelines and operationalized in National nurses’ regulatory body and curricular for nurses in Ghana.

Response: We have attempted to address the issues raised here.

Comment:

There is also need to describe how the ethical guidelines for nurses relate to the specific areas of Ghana constitution.

Response: We made changes to the literature reflecting these issues.

Comment:

My general view is that assessment of knowledge, attitude and practice of nurses in the area of ethics is an important issue but the manuscript needs to be revised substantially before publication as follows:

Introduction: There is need for major revision to give description in the subject matter of ethics, magnitude of the problem, the negative effects and gaps in Knowledge, attitudes and practice that justify the study.

Response: We have made changes to improve the introductory part of the manuscript.

Comment:

The literature review does not demonstrate logical arguments that justify the study

Response: We have made changes to improve the literature.

Comment:

Methodology: There are major methodological issues of concern indicated on the specific areas (see comments in the manuscripts for detailed suggestions).

Response: We have made substantive revision to the method section to reflect the comments made by the reviewers.

Comment:

Results and Discussion: Need to be revised as indicated in the comments within the document.

Generally-there is need to revise the manuscript for logical flow.

Response: We have revised the results and discussion for logical flow in the manuscript.

Comment:

Declaration of competing interest-I declare that I do not have any conflicting interest

Response: We added that to the manuscript. 

Reviewer #2: 

Comment:

This is an interesting and informative paper on a geographically understudied area of nursing practice. As I am a qualitative researcher, I cannot comment on the regression analysis specifically. Outside of that, the study design and methodology seem sound and the results support the conclusions, with one caveat:

Response: We appreciate your comments.

Comment:

I would encourage the authors to consider and discuss the difference between 1) 'ethics' in a general social sense, 2) professional ethics as specific to nursing, and 3) the particular written national codes of ethics they discuss in the paper. They should make clear that when discussing 'nurse's knowledge of ethics', they mean the third sense, namely formal written codes of conduct. I'm saying this because it could be well possible that rather than having 'no ethics at all', as the authors seem to imply, the nurses in question simply operate on a different (maybe not formal/bureaucratic) understanding of ethics, which then conflicts with professional standards. For example, their treatment of women in antenatal clinics may not be in tune with formal ethical standards at the hospital - but is it possible that it conforms to a 'local' ethics which demands that women attend the clinic and pay for the items needed there? These distinctions might make for an interesting follow-up study, but as it stands, the data does not support the claim that nurses do not have any knowledge of any ethics at all - just that they do not have knowledge of the relevant formal codes applying to their jobs.

Response: We appreciate the comments. We made changes to reflect ethical practices of nurses in the Ghanaian context. 

Comment:

I would also recommend to have the article professionally edited if at all possible, to eliminate potential misunderstandings due to language use (e.g. p 57 ln 59 - 'couples'?).

Response: We deleted the word “couples” to make the sentence readable.

Comment:

The authors state that the original data will be made available upon request, but it appears that PLOS ONE requires data to be deposited in a public depository for the paper to be accepted. Once these points are addressed, I warmly recommend the paper for publication.

Response: We are making the necessary efforts to upload the data.

Reviewer #3:

Comment:

1. Page Number - PN (3) Line Number - LN (58): - Is this issue of unethical practice among nurses a global one or just specific to some places? Or are there other forms in other places?

Response: We have attempted to rewrite the sentence for clarity.

Comment:

2. PN (3) LN (75-79): - Where were these studies conducted? Is this a global problem or localized? You need to be specific here. There might be a need to present existing global studies especially from the western world in relation to the unethical nursing practices. LN 79 - Please change to ‘Midwifery’.

Response: We have rewritten the sentences to make better understanding of the issues raised.

Comment:

3. PN (4), LN (85-86): - This sentence is not clear. How does the attitude and the practice of nurses influence how they act and conduct their duties? Or that the nurses’ knowledge of the ethics of their profession influenced their attitudes and practices? LN (95-97): - Is this referring to the nurses in Ghana?

Response: We have improved the literature. 

Comment:

4. PN (5), LN (116): - Change to ‘bachelor’s degree’

Response: We made the change accordingly. 

Comment:

5. PN (6), LN (144-147): - This appears confusing. Can you please elaborate on this sentence especially regarding the evaluation by the two senior research assistants? Was this part of the analysis? LN (149-150): This is not clear. Which of the group of nurses did not take part in the main study? Why were they not involved in the main study? Was there a pilot test?

Response: We have rewritten this part for clarity. This was a pilot test of the questionnaire. 

Comment:

6. PN (10), LN (273): - Change to ‘…and in Ghana’ LN (294): Please rewrite this.

Response: We made the change accordingly.

Comment:

7. PN (11), LN (305-307): - Yes, this is true. However, do the nurses in Ghana or any of the African countries face lawsuits for unethical practices? Are the people aware of their rights regarding the unethical practices of the nurses who take care of them? Have you also considered the issues of quackery in the nursing profession, whereby people assumed that everyone working in health facilities wearing ‘white uniforms’ are nurses? In essence, what role does quackery play in the unethical practices observed among the nurses in Ghana?

Response: We have made the necessary improvement to this part of the discission. 

Comment:

LN (314 - 320): From your finding, and with the high percentage of ethical practices among nurses in Ghana, what could be the explanation for these inhibiting factors raised here and the high percentage of ethical practices among the nurses interviewed in this study?

Response: We have made modifications to this.

Comment:

8. PN (12), LN (332-333): - You need to expatiate more on this issue. Are diplomate nurses found to be more involved in unethical practices as a result of their low level of education? Or what?

From this study findings, nurses who have diploma in nursing also have high level of knowledge about the ethics of nursing profession. This appears to contradict earlier studies in relation to this.

Response: We have expatiated this appropriately.

Comment:

LN (349-350): How can the findings from this study be generalized to other health care professionals? Are they also influenced by the same factors? Are there studies that suggest that they also practice unethically? Which of the health care professionals do you want to generalize your findings to?

Response: We have restated this section appropriately to reflect that we refereeing to only nurses.

Comment:

LN (350-353): Further study should be encouraged to identify specific nursing specialties where there are reports of unethical practices. There is also a need to identify the specific aspects of the nursing ethics where unethical practices have been identified among nurses. Studies should be done to identify if there are complaints about unethical practices among other health care professionals.

Response: We have modified this appropriately.

Comment:

9. PN (13), LN (357-358): There should be sanctions and penalties for nurses who are reported and confirmed to be involved in unethical practices.

Response: We added a sentence that calls for sanctions and penalties for nurses with unethical practices.

---

## [Decision Letter · Decision Letter 1]

17 Aug 2021

PONE-D-20-10854R1

Ethics in healthcare: Knowledge, attitude and practices of nurses in the Cape Coast Metropolis of Ghana

PLOS ONE

Dear Dr. Sambah,

Thank you for submitting your manuscript to PLOS ONE. After careful consideration, we feel that it has merit but does not fully meet PLOS ONE’s publication criteria as it currently stands. Therefore, we invite you to submit a revised version of the manuscript that addresses the points raised during the review process.

A 'Response to Reviewers' letter that responds to each point raised by the academic editor and reviewer(s). You should upload this letter as a separate file labeled 'Response to Reviewers'.A marked-up copy of your manuscript that highlights changes made to the original version. You should upload this as a separate file labeled 'Revised Manuscript with Track Changes'.An unmarked version of your revised paper without tracked changes. You should upload this as a separate file labeled 'Manuscript'.

We look forward to receiving your revised manuscript.

Kind regards,

Prof. Ritesh G. Menezes, M.B.B.S., M.D., Diplomate N.B.

Academic Editor

PLOS ONE

Reviewers' comments:

Reviewer's Responses to Questions

**Comments to the Author**

1. If the authors have adequately addressed your comments raised in a previous round of review and you feel that this manuscript is now acceptable for publication, you may indicate that here to bypass the “Comments to the Author” section, enter your conflict of interest statement in the “Confidential to Editor” section, and submit your "Accept" recommendation.

Reviewer #1: All comments have been addressed

Reviewer #4: (No Response)

2. Is the manuscript technically sound, and do the data support the conclusions?

Reviewer #1: (No Response)

Reviewer #4: Partly

3. Has the statistical analysis been performed appropriately and rigorously? 

Reviewer #1: (No Response)

Reviewer #4: No

4. Have the authors made all data underlying the findings in their manuscript fully available?

Reviewer #1: (No Response)

Reviewer #4: Yes

5. Is the manuscript presented in an intelligible fashion and written in standard English?

Reviewer #1: (No Response)

Reviewer #4: No

6. Review Comments to the Author

Reviewer #1: (No Response)

Reviewer #4: • line 34: Method – survey design is not appropriate term. it is preferable to use cross-sectional design

• line 38: high ethical knowledge is not appropriate – it is better to state good knowledge

• line 39: 98% were ethical conscious in their practices – it is better to state 98% had good ethical practices

• line 44: nurses’ practice with high ethical standards – needs to be rephrased

• MeSh terms to be used for Keywords.

• lines 78-80: what is the reference /basis for these statements?

• lines 95-96: rephrase statement

• lines 90-115: sentences look repetitive. they may be condensed. syntax errors to be corrected. it seems that problems/concerns with nurse patient communication have been interchangeably used with ethical behaviour. This is not appropriate & needs to be corrected.

• line 119: how was sample size calculated?

• line 130: of the 42 items – only details of 30 items have been mentioned in section A & B

• lines 164-168: grammatical corrections needed

• lines 190-192: how were knowledge, attitude & practice scores aggregated as continuous scores? why were the variables not analysed as categorical variables?

• line 193: what is a latent variable?

• line 194: what is mixture of variables? – it is preferable to utilize appropriate statistical terms.

• line 195: p value<0.05

• what software was used for data analysis?

• line 200: high knowledge – to be corrected

• line 208-209: table title needs to be reworded. level of ethical knowledge is incorrect usage

• tables can be reframed to depict key variables rather than reproducing the data output tables

• line 225: CHN/RCN/RMN – the explanation for the abbreviated terms need to be provided as footnotes under the table

• lines 230-232: it is preferable to classify knowledge as good & poor

• line 311: high level of ethical knowledge to be replaced by good knowledge about ethics

• line 325: who could not be a presentation of the all nurses in Ghana. – needs to be rephrased as not representative of all nurses in Ghana

• lines 331-332: sentences to be rephrased

• lines 334-337: please substantiate your claim on attitudes & patient recovery

• lines 357-358: syntax error to be corrected

• lines 368-370: sentences need to be rephrased

• line 379: conclusion- should summarize key findings of the study in 3-4 sentences

• lines 380-383: may be removed from conclusions

• lines 384-388 avoid usage of term – high ethical standards

• lines 395-396: sentences need to be rephrased

• lines 391-393: are limitations of the study & not conclusions

• line 414: syntax error

• the tool/questionnaire used for the study may be enclosed for better clarity on the questions

7. PLOS authors have the option to publish the peer review history of their article (what does this mean?). If published, this will include your full peer review and any attached files.

Reviewer #1: No

Reviewer #4: No

---

## [Author Response · Author response to Decision Letter 1]

14 Sep 2021

Response to Editor and Reviewers comments

“line 34: Method – survey design is not appropriate term. it is preferable to use cross-sectional design:

Response: We have made the correction as appropriate (line 34). 

• “line 38: high ethical knowledge is not appropriate – it is better to state good knowledge”

Response: We changed it to the “good knowledge” (line 38) 

• “line 39: 98% were ethical conscious in their practices – it is better to state 98% had good ethical practices”

Response: We corrected the sentence in (line 39)

• “line 44: nurses’ practice with high ethical standards – needs to be rephrased” 

Response: We rephrased the sentence as appropriate (lines 44&45)

• “lines 78-80: What is the reference /basis for these statements?”

Response: We provided the references to these statements

• “lines 95-96: rephrase statement”

Response: We corrected the statement 

• “lines 90-115: Sentences look repetitive. They may be condensed. syntax errors to be corrected. it seems that problems/concerns with nurse patient communication have been interchangeably used with ethical behaviour. This is not appropriate & needs to be corrected.”

Response: we restructured these sentences 

• “line 119: How was sample size calculated?”

Response: We used all the nurses in the three facilities, which we described under the method section. 

• “line 130: of the 42 items – only details of 30 items have been mentioned in section A & B”

Response: we actually used 32 items, which are accounted for in the descriptions 

• “lines 164-168: grammatical corrections needed”

Response: we provided the corrections as needed 

• “lines 190-192: how were knowledge, attitude & practice scores aggregated as continuous scores? why were the variables not analysed as categorical variables?”

Response: These explanations are provided under the data analysis (lines 168-185)

• “line 193: what is a latent variable?”

Response: We did provide a clear description for this.

• “line 194: what is mixture of variables? – it is preferable to utilize appropriate statistical terms.”

Response: We provided an appropriate description for that.

• “line 195: p value<0.05”

Response: We made the change as required.

• “what software was used for data analysis?”

Response: The correction has been made, as we used SPSS software, version 20.0.

• “line 200: high knowledge – to be corrected”

Response: We change it to good knowledge

• “line 208-209: table title needs to be reworded. level of ethical knowledge is incorrect usage. tables can be reframed to depict key variables rather than reproducing the data output tables”

Response: We change the table tittle as suggested 

• “line 225: CHN/RCN/RMN – the explanation for the abbreviated terms need to be provided as footnotes under the table”

Response: We provided the full meaning of the abbreviations 

• “lines 230-232: it is preferable to classify knowledge as good & poor”

Response: We change the phrases good and poor knowledge 

• “line 311: high level of ethical knowledge to be replaced by good knowledge about ethics”

Response: We changed the sentence to reflect “goof knowledge”

• “line 325: who could not be a presentation of the all nurses in Ghana. – needs to be rephrased as not representative of all nurses in Ghana”

Response: This sentence has been corrected appropriately

• “lines 331-332: sentences to be rephrased”

Response: Sentences have been corrected 

• “lines 334-337: please substantiate your claim on attitudes & patient recovery”

Response: We provide references and examples in support of the claims

• “lines 357-358: syntax error to be corrected”

Response: The correction was made

• “lines 368-370: sentences need to be rephrased”

Response: Sentences have been corrected 

• “line 379: conclusion- should summarize key findings of the study in 3-4 sentences”

Response: We summarized the key findings appropriately 

• “lines 380-383: may be removed from conclusions”

Response: We removed the sentences as suggested.

• “lines 384-388 avoid usage of term – high ethical standards”

Response: We change to good knowledge 

• “lines 395-396: sentences need to be rephrased?”

Response: Sentences have been corrected 

• “lines 391-393: are limitations of the study & not conclusions”

Response: These sentences have been removed 

• “line 414: syntax error”

Response: We did correct this sentence

---

## [Decision Letter · Decision Letter 2]

1 Nov 2021

PONE-D-20-10854R2

Ethics in healthcare: Knowledge, attitude and practices of nurses in the Cape Coast Metropolis of Ghana

PLOS ONE

Dear Dr. Sambah,

Thank you for submitting your manuscript to PLOS ONE. After careful consideration, we feel that it has merit but does not fully meet PLOS ONE’s publication criteria as it currently stands. Therefore, we invite you to submit a revised version of the manuscript that addresses the points raised during the review process.

Please submit your revised manuscript by December 16, 2021. Please include the following items when submitting your revised manuscript:

A 'Response to Reviewers' letter that responds to each point raised by the academic editor and reviewer(s). You should upload this letter as a separate file labeled 'Response to Reviewers'.A marked-up copy of your manuscript that highlights changes made to the original version. You should upload this as a separate file labeled 'Revised Manuscript with Track Changes'.An unmarked version of your revised paper without tracked changes. You should upload this as a separate file labeled 'Manuscript'.

We look forward to receiving your revised manuscript.

Kind regards,

Prof. Ritesh G. Menezes, M.B.B.S., M.D., Diplomate N.B.

Academic Editor

PLOS ONE

Journal Requirements:

Reviewers' comments:

Reviewer's Responses to Questions

**Comments to the Author**

1. If the authors have adequately addressed your comments raised in a previous round of review and you feel that this manuscript is now acceptable for publication, you may indicate that here to bypass the “Comments to the Author” section, enter your conflict of interest statement in the “Confidential to Editor” section, and submit your "Accept" recommendation.

Reviewer #4: All comments have been addressed

2. Is the manuscript technically sound, and do the data support the conclusions?

Reviewer #4: Yes

3. Has the statistical analysis been performed appropriately and rigorously? 

Reviewer #4: Yes

4. Have the authors made all data underlying the findings in their manuscript fully available?

Reviewer #4: Yes

5. Is the manuscript presented in an intelligible fashion and written in standard English?

Reviewer #4: No

6. Review Comments to the Author

Reviewer #4: Line 44: Good high ethical standards is not appropriate phrase. Please correct the statement

Line103: improperly carried out ethical practices in their routine care like insulting patient – please rephrase

Lines 125-126: It is preferable to state that all the 326 nurses working in the municipality hospitals were included in the study

Table 2,3,4: titles can be modified to improve understanding

Line 330: The finding again indicates a high positive attitude – avoid usage of phrases like high positive attitude

Line 333: Oceanian – spell check for errors

Lines 369-372: sentences are grammatically incorrect. Kindly rephrase

7. PLOS authors have the option to publish the peer review history of their article (what does this mean?). If published, this will include your full peer review and any attached files.

Reviewer #4: No

---

## [Author Response · Author response to Decision Letter 2]

21 Dec 2021

Respond to Comments from the Reviewer 

We are very grateful to reviewers for the thorough review done so far on our paper. We provide the following responses to the comments. We indicated our corrections in Red Link in the main document (Revised Manuscript with Track Changes). 

1. Updated references

Response:

2. Line 44: Good high ethical standards is not appropriate phrase. Please correct the statement.

Response: We corrected this, by deleting “high”.

3. Line 103: Improperly carried out ethical practices in their routine care like insulting patients. Please rephrase.

Response: we corrected this sentence, lines 102-103

4. Lines 125-126: It is preferable to state that all the 326 nurses working in the municipality hospitals were included in the study. 

Response: This sentence is changed accordingly, line 124

5. Tables 2, 3, 4: Titles can be modified to improve understanding.

Response: Titles have been modified accordingly, lines 258-259, 276-277, 292-293

6. Line 330: The finding again indicates a high positive attitude- avoid usage of phrases like high positive attitude.

Response: We introduced “good” attitude in the sentence, line 324

7. Line 333: Oceanian – spell check for error.

Response: We correct, “Oceania”, line 327

8. Lines 369-372: Sentences are grammatically incorrect. Kindly rephrase.

Response: We attempted to correct the sentences for understanding, lines 362-364

---

## [Editor Report · Decision Letter 3]

24 Jan 2022

Ethics in healthcare: Knowledge, attitude and practices of nurses in the Cape Coast Metropolis of Ghana

PONE-D-20-10854R3

Dear Dr. Sambah,

We’re pleased to inform you that your manuscript has been judged scientifically suitable for publication and will be formally accepted for publication once it meets all outstanding technical requirements including language corrections.

Within one week, you’ll receive an e-mail detailing the required amendments. You must seek independent editorial help to address language issues. Refer to the 5th criterion for publication listed by PLoS ONE (https://journals.plos.org/plosone/s/criteria-for-publication). When these have been addressed, you’ll receive a formal acceptance letter and your manuscript will be scheduled for publication. 

Kind regards,

Prof. Ritesh G. Menezes, M.B.B.S., M.D., Diplomate N.B.

Academic Editor

PLOS ONE

Additional Academic Editor Comments:

Not all grammatical mistakes are addressed by the authors despite providing 3 opportunities to revise the manuscript. PLoS ONE does not copy edit accepted manuscripts. Therefore, authors must address language issues preferably while addressing other technical requirements or at least at the time of proof corrections.

---

## [Editor Report · Acceptance letter]

7 Feb 2022

PONE-D-20-10854R3 

Ethics in Healthcare: Knowledge, Attitude and Practices of Nurses in the Cape Coast Metropolis of Ghana 

Dear Dr. Sambah:

I'm pleased to inform you that your manuscript has been deemed suitable for publication in PLOS ONE. Congratulations! Your manuscript is now with our production department. 

Kind regards, 

on behalf of

Prof. Dr. Ritesh G. Menezes 

Academic Editor

PLOS ONE